# *Phyllanthus* Lignans: A Review of Biological Activity and Elicitation

**Winda Nawfetrias** [1,2], **Lukita Devy** [2], **Rizkita Rachmi Esyanti** [1] **and Ahmad Faizal** [1,*]

[1] Plant Science and Biotechnology Research Group, School of Life Sciences and Technology, Institut Teknologi Bandung, Bandung 40132, Indonesia; winda.nawfetrias@brin.go.id (W.N.); rizkita@itb.ac.id (R.R.E.)

[2] Research Center for Horticulture, Research Organization for Agriculture and Food, National Research and Innovation Agency, Cibinong Science Center, Bogor 16915, Indonesia; lukita.devy@brin.go.id

[*] Correspondence: afaizal@itb.ac.id

**Abstract:** The *Phyllanthus* genus exhibits a broad distribution spanning across the majority of tropical and subtropical regions. Due to their ability to synthesize medicinal bioactive compounds such as lignans, they have been utilized historically in traditional medicine to treat a wide range of ailments. This review discusses the current knowledge on the potency of lignans for medicinal purposes, the benefit of lignans for plants, various lignans produced by *Phyllanthus*, and how lignan synthesis could be increased through biotic and abiotic elicitation. Finally, we present a set of connected hypotheses to explain how signaling crosstalk between endophytic microbes and drought stress responses regulates lignan production. Although the mechanisms of lignan synthesis in *Phyllanthus* are not fully explored, this review strongly supports the view that endophytic fungi and drought stress can increase lignan production in plants belonging to the genus *Phyllanthus*. The medicinal plant–endophyte–drought stress relationship helps to improve the lignan yield of *Phyllanthus*, which is crucial for human health and can be optimized under in vitro and in vivo conditions.

**Keywords:** drought stress; elicitation; endophytic fungi; lignan; *Phyllanthus*

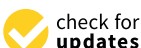



## 1. Introduction

Plants have traditionally been essential in treating various human diseases and ailments. Extracts and decoctions made from plants are commonly used in traditional medicine. Research to develop novel pharmaceutical formulations has often included medicinal plants as an important component. Because synthetic pharmaceuticals can have negative effects and their overuse can result in drug resistance and other challenges, there is enormous interest in exploring medicinal plants for natural and alternative medicines. Medicinal plants remain the main source of healthcare for the majority of people globally and have been a staple in traditional health systems for many years [1,2].

One of the reasons that medicines derived from plants have been so successful is that they generally cause fewer adverse side effects than synthetic treatments. According to the World Health Organization, data indicate that plants are effective in treating diseases for 80% of the global population, and the rate is higher in developing countries than in developed countries [3]. In many countries, plant resources are less expensive and more accessible than synthetic medicines, making medicinal plants preferable to synthetic pharmaceuticals [4].

Approximately 70,000 plant species are known to be utilized for the treatment of diseases, but only about 15% of the plant species growing in the world have been researched for their medicinal value. Despite this low percentage, 25% of conventional medicines used in contemporary medicine are derived from plants [2,5]. The National Basic Health Survey shows that 40–59% of the population uses indigenous traditional medicine practices and herbal medicines in Indonesia. Regulations apply to acupuncture, chiropractic, and herbal medicine providers at national, state, and local government levels [6]. Plants widely

used for medicinal purposes belong to the families Phyllantacheae [7], Euphorbiacea, and Zingiberaceae [8].

*Phyllanthus* is a genus within the Phyllanthaceae family, encompassing 1314 species. It is extensively found across tropical and subtropical areas in the world. The most significant distributions of *Phyllanthus* are found in Australia, Brazil, the United States, China, Mexico, New Caledonia, China Taipei, Colombia, Reunion, and Indonesia. Several species within *Phyllanthus* are recognized as herbaceous plants, including *Phyllanthus amarus*, *Phyllanthus urinaria*, *Phyllanthus odontadenius*, and *Phyllanthus niruroides* (Figure 1). However, *P. muellerianus* is notable for its shrubby growth form. These plants typically reach heights ranging from 60 to 100 cm. Their leaves are characterized by their elliptic–oblong shape, measuring 4–10 mm in length and 1.5–7 mm in width. The leaf tip can be either obtuse or mucronate. The stems are greenish, smooth, rounded, glabrous, and exhibit a woody texture at the base. The fruits vary in color, presenting as greenish, spotted-green, or reddish, and are accompanied by tepals that can be reddish-green or whitish-green, depending on the specific species. Tepals are usually groups of five or six. The stipules are typically greenish-reddish and are positioned laterally in a free manner. Distinct from the others, *P. muellerianus* features leaves that are 2–7 cm in length and 1.5–4 cm wide, with an ovate–elliptic shape and subacute apex. The stems of this species are brownish-green, thorny, rounded pentagonal, glabrous, and completely woody. The fruits resemble red berries, while tepals are greenish. Like other species in the genus, *P. muellerianus* has five tepals. Its stipules are greenish and spiny [9].

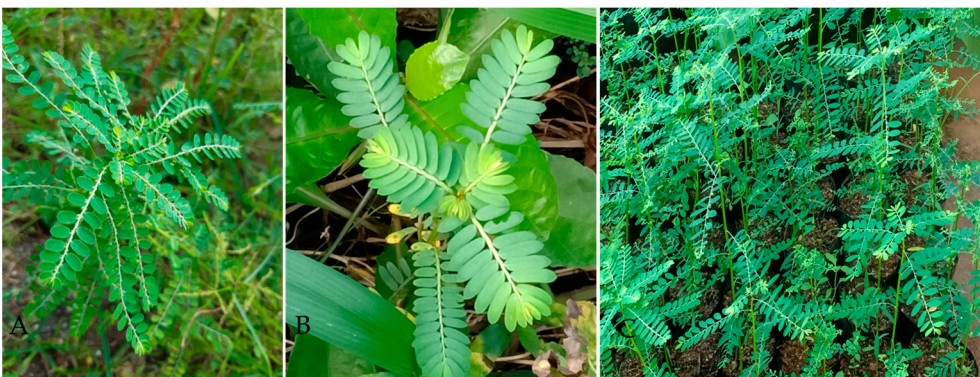

**Figure 1.** Various species of the genus of *Phyllanthus*: (**A**) *P. amarus*, (**B**) *P. urinaria*, and (**C**) *P. niruri*.

The *Phyllanthus* genus, known for its role in traditional medicine, is used to treat a variety of ailments, such as kidney and bladder problems, intestinal infections, diabetes, and hepatitis B. Among its species, several have been studied for their chemical and medicinal properties. These include *P. niruri*, *P. accuminatus*, *P. amarus*, *P. niruroides*, *P. anisolobus*, *P. emblica*, *P. oxyphyllus*, *P. flexuosus*, *P. raticulatus*, *P. fraternes*, *P. simplex*, *P. mullernus*, *P. urinaria*, *P. mytrifolis*, *P. virgatus*, *P. orbiculatus*, *P. pulcher*, and *P. watsonii* [10]. In recent years, *Phyllanthus* has received a significant amount of attention as a result of several factors, including (1) its widespread availability in many tropical and subtropical countries, (2) its large number of constituent species, (3) its widespread therapeutic applications in traditional medicines, and (4) the greater variety of specialized metabolites that can generally be found in plants. As a result, in vivo and in vitro studies and a few clinical investigations have made significant achievements in uncovering the chemical and pharmacological properties of certain *Phyllanthus* species [11].

*Phyllanthus* plants contain secondary metabolites or bioactive compounds, including phenolics, terpenoids, alkaloids, anthocyanins, chlorogenic acids, flavonoids, tannins, glycosidic substitutes, and lignans [12]. Among these, lignans are a significant class of natural compounds known for their health benefits arising from their antioxidant, anticarcinogenic, antimutagenic, and anti-estrogenic properties. These compounds are synthesized via the shikimic acid pathway and composed of dimerized phenylpropanoid units. This results in

a structure exhibiting varied oxidation levels and substitution patterns on their aromatic moiety [13]. Specifically, the structure of lignan is formed through the dimerization of the two central carbon atoms (8 and 8') in the side chain of the phenylpropanoid unit, which has a C6C3 configuration [14] (Figure 2).

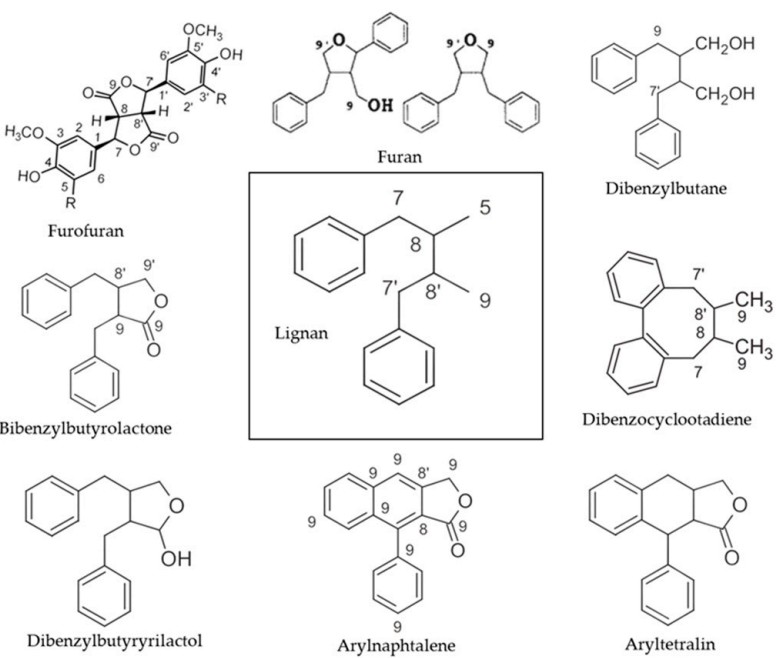

**Figure 2.** Basic skeletons of lignan and lignan subgroups [15].

Spectrometry analysis techniques such as Thin Layer Chromatography (TLC), High-Performance Liquid Chromatography (HPLC), Liquid Chromatography–Mass Spectrometry (LC-MS), and Gas Chromatography–Mass Spectrometry (GC-MS) play crucial roles in the identification of lignans in *Phyllanthus* species. Among these, TLC is known for its simplicity and cost-effectiveness in separating and identifying compounds within mixtures. This method, while useful for analyzing lignans in *Phyllanthus* species, lacks the sensitivity and selectivity offered by more advanced techniques like HPLC and LC-MS [16].

HPLC stands out as a robust tool for separating and identifying lignans in various *Phyllanthus* species, offering high resolution for different components. This technique has been effectively applied to the study of lignans in species such as *P. urinaria* and *P. niruri* [16,17]. Furthermore, HPLC can be enhanced through integration with Solid-Phase Extraction–Nuclear Magnetic Resonance (SPE-NMR), which aids in more efficient and definitive identification of lignans.

LC-MS, which combines the separation of liquid chromatography with the analytical power of mass spectrometry, excels in terms of sensitivity and specificity for identifying lignans in *Phyllanthus* species. This method has been employed in the analysis of lignans from *P. urinaria* and *P. niruri*, yielding clear 1H-NMR spectra for the separated components [16,17].

GC-MS, another valuable technique, distinguishes and identifies compounds based on their volatility and mass-to-charge ratios. While specific examples of its use in analyzing lignans in *Phyllanthus* species have not been detailed in the referenced material, its general application in this field is acknowledged.

Lignans are usually found in small amounts in plants. Their synthesis is one of the main issues that must be addressed to accomplish further bioactivity studies [13]. Two approaches to enhance the production of secondary metabolites in plants include the addition of biosynthetic precursors and the use of elicitors. Biotic and abiotic elicitors can boost the yield of phytochemicals by triggering a defense response in plants, akin to the reaction observed during a pathogen attack or under stressful environmental conditions [18]. Elici-

tors have been extensively studied for their ability to increase productivity and secondary metabolite production in plants [19]. This review aims to compile the current research on *Phyllanthus* and critically evaluate the issues regarding *Phyllanthus* phytochemistry, pharmacology, and elicitation, especially for lignan compounds.

The data compilation for this study primarily utilized databases such as Google Scholar, Scopus, Science Direct, and Elsevier, encompassing publications from 1936 to 2024. The research focused on specific keywords: "lignan", "Phyllanthus", "elicitation", "endophyte" and "drought". The criteria for selecting research papers were stringent to ensure relevance and quality. Papers included in this study needed to be original research articles related to *Phyllantus*. Key areas of focus were the mechanisms of action in elicitation processes induced by endophytic fungi and drought conditions, both biotic or abiotic factors influencing lignan compounds in *Phyllantus*. Additionally, studies that explored extracts from various *Phyllanthus* species and their contribution to biological activities were considered. This research encompassed diverse study designs, including in vivo, in vitro, and ex vivo approaches, to provide a comprehensive understanding of the subject.

## 2. *Phyllanthus* for Medicinal Purposes

*Phyllanthus* spp. Is known for its potency as a medicinal plant. The herb *P. niruri* is widely used to treat hepatic disease, edema, dropsical conditions, and urinary problems [20]. Several bioactive components can be obtained in *P. niruri* extract, including lignans such as hypophyllanthin, phyllanthin, and phyltetralin [12]. *P. niruri* is utilized in different regions globally, particularly in Asia and America, for its diuretic, antiviral, anti-inflammatory, and hepatoprotective properties, among other effects [21]. Ethnobotanical studies of native Indonesian medicine reported that *P. niruri* is used to treat herpes zoster in Dayak Ngaju tradition [22], appendicitis in the Seko tribe of Central Sulawesi [23], and postpartum discomfort in the Enggano tribe [24]. *P. urinaria*, another herb species in the *Phyllanthus* genus, displays antiviral, antitumor, hepatoprotective, antidiabetic, antioxidant, antihypertensive, anti-inflammatory, and antimicrobial effects. Phytochemical analysis of *P. urinaria* extracts has identified the presence of several compounds, including lignans, coumarins, and flavonoids [25]. *P. emblica* has a long history of use in various traditional medical systems around the world, owing to its remarkable healing and rejuvenating properties. This has garnered attention from the biopharmaceutical industry for potential use as an alternative medicine to treat many illnesses [26]. *P. amarus*, rich in compounds like polyphenols, flavonoids, lignans, triterpenes, sterols, tannins, and alkaloids, is a key component in Indian traditional Ayurvedic medicine. It is used for a variety of treatments, including as a diuretic, fever reducer, and antiseptic and for addressing issues related to the stomach, genitourinary system, liver, kidneys, and spleen. Additionally, it is used for treating gonorrhea, gastropathy, diarrhea, ophthalmopathy, ulcers, and wounds [27].

The pharmacological potency of *Phyllanthus* stems from its capacity to produce a range of bioactive compounds that are beneficial for human use. Saponins, flavonoids, steroids, alkaloids, phenols, terpenoids, tannins, glycosides, 5,6,7-trimethoxyflavone, and apigenin-8-C-glucoside were identified by LC-MS analysis of ethanolic *P. niruri* leaf extract [28]. In *P. niruri*, the presence of catechin and quercetin contributes to a reduction in TNF-, IL-1, IL-6, and iNOS expression. This action helps suppress the inflammatory response and serves as an agent that modulates the immune system [29]. Ethanolic extracts from the leaves of *P. niruri* contain saponin and terpenoids with strong butyrylcholinesterase inhibitory action, which is of potential interest in Alzheimer's treatment [30]. Substances such as quercetin, kaempferol, and other flavonoids found in *Phyllanthus* demonstrate potential in inhibiting the enzyme xanthine oxidase. The inhibition of xanthine oxidase is enhanced by hydroxyl groups on flavonoid compounds. Additionally, the interactions involve the planar rings A and C of flavones, a subclass of flavonoids, with phenylalanine residues at positions 1009 and 914 of the enzymes. Xanthine oxidase is essential for the breakdown of purines, compounds present in various foods. It facilitates the oxidation of hypoxanthine to xanthine and subsequently, xanthine to uric acid. Elevated levels of

uric acid in the blood can lead to the development of kidney stones and renal failure. This enzyme also plays a role in various diseases, including cardiovascular diseases and cancer [31]. The high binding affinity from natural substances found in *P. amarus* such as corilagin, furosin, geraniin, and other flavonoids (reaching up to 10.60 kcal/mol) compared to remdesivir (with a maximum of 9.50 kcal/mol) opens new avenues for the treatment of SARS-CoV-2 infection using plant-based compounds [32].

## 3. Lignans in Plants

Lignans, lignin, tannins, flavonoids, and stilbenes comprise a class of phenolic derivatives that play crucial roles in plants. Although they have similar biochemical bases, different lignans have different structures and biological activities [33]. Lignans have been found in various parts of plants, including flowers, seeds, leaves, roots, fruits, and woods. Lignans mostly act as defense mechanisms but have other functions [34,35].

Lignans help plants to survive under biotic and abiotic stress conditions and are crucial in the defense strategies of plants against various pests or diseases [36]. For example, the lignan yatein protects tomato plants from the fungus Botrytis cinerea by preventing the fungus' growth [37]. Sesame plants with Fusarium infections have more substantial accumulations of some lignans than healthy plants, suggesting that accumulated lignans in sesame have an antioxidant impact by reducing reactive oxygen species (ROS) production and protecting infected plants against oxidative stress [38]. Lignans are potential antioxidants because of their structure, which enables them to scavenge ROS accumulated under stressful conditions. These compounds act as natural antioxidants by scavenging free radicals produced during oxidation reactions, which occur naturally, mostly associated with metabolic pathways [39,40] and the regulation of plant development [41,42].

An integrated study of transcriptome and metabolic profiles revealed the correlation between lignan production and biomass growth on salt and drought resistance [43]. Dirigent proteins (DIRs) and peroxidases regulate lignification levels when plants are subjected to abiotic stress. In contrast, during lignan biosynthesis, DIRs are crucial mediators of regio- and stereo-selective bimolecular coupling [44].

Lignans can also be produced by fungi and bacteria called endophytes, which live in plant tissues without causing visible signs of disease. Endophytes are capable of showing some of the same therapeutic properties as plants and can also secrete known or novel metabolites [45]. Several endophytic fungi produce lignan compounds such as aryl naphthalene lignan podophyllotoxin [13]. Edenia gomezpompae, an endophytic fungus from *P. amarus*, can synthesize lignans such as phyllanthin [46]. The endophytic fungus Trametes hirusta isolated from the rhizomes of *P. hexandrum* produced aryl tetralin lignan [47]. The endophytic fungus *Nigrospora* sp., found in *P. amarus*, has been reported to produce phyllanthin, hypophyllanthin, and other antioxidant compounds [48]. The endophytic fungus Aspergillus niger obtained from the aerial parts of *P. amarus* exhibited antihepatotoxic activity, indicating its potential for liver protection.

## 4. Lignans from *Phyllanthus*

Natural lignans can be further developed as the main compounds in new antiviral medicines [49]. Research has identified numerous lignans with antiviral properties across a wide range of plant species (66), genera (43), and families (34). Particularly noteworthy are the lignans found in the *Kadsura*, *Schisandra*, and *Phyllanthus* genera, which have shown promising anti-human immunodeficiency virus (HIV) and anti-human hepatitis B virus (HBV) effects [13,50,51]. The specific lignans discovered in *Phyllanthus* are provided in Table 1.

**Table 1.** A detailed list of various lignans that have been identified within the genus *Phyllanthus*.

| Lignan | Structure | Species | Refs. |
|---|---|---|---|
| Phyllanthin $C_{24}H_{34}O_6$ | | *P. niruri* *P. amarus* *P. tenellus* *P. urinaria* | [49,52–54] |
| Hypophyllanthin $C_{24}H_{30}O_7$ | | *P. niruri* *P. amarus* *P. tenellus* *P. urinaria* | |
| Phylltetralin $C_{24}H_{32}O_6$ | | *P. niruri* *P. tenellus* *P. urinaria* | |
| Niranthin $C_{24}H_{32}O_7$ | | *P. niruri* *P. amarus* *P. tenellus* *P. urinaria* | |
| Isolintetralin $C_{23}H_{28}O_6$ | | *P. amarus* *P. urinaria* | |
| Nirtetralin $C_{24}H_{30}O_7$ | | *P. urinaria* | |
| Lintetralin $C_{23}H_{28}O_6$ | | *P. urinaria* | |
| Heliobuphthalmin lactone $C_{20}H_{18}O_6$ | | *P. urinaria* | |
| Virgatusin $C_{23}H_{28}O_7$ | | *P. urinaria* | |
| Urinatetralin $C_{22}H_{24}O_6$ | | *P. urinaria* | |
| Dextrobursehernin $C_{21}H_{22}O_6$ | | *P. urinaria* | |
| Urinaligran $C_{22}H_{24}O_7$ | | *P. urinaria* | |

The methanolic extract derived from various parts of *P. amarus* includes six lignans: isolintetralin, demethylenedioxy-niranthin, niranthin, phyllanthin, and hypophyllanthin, as well as a triterpene identified as 2Z, 6Z, 10Z, 14E, 18E, 22E-farnesil farnesol [52]. Phyllamycin B and retrojusticidin B, lignans synthesized by *P. myrtifolius*, are effective in inhibiting the activity of HIV-1 reverse transcriptase [50]. The lignans hinokinin and niranthin from *Phyllanthus* could be dissolved in dimethyl sulphoxide (DMSO) and were found to inhibit HBV replication by 33.9% and 68.3%, respectively [49]. Furthermore, the lignans phyllanthin and hypophyllanthin from *Phyllanthus* species were found to have liver-protecting properties [55]. In the Indonesian Herbal Pharmacopoeia, these lignans have been identified as chemicals used as benchmarks for ensuring the quality of *Phyllanthus* plants [56]. The lignan phyllanthin also serves a significant role in plant defense mechanism due to its radioprotective and free radical scavenging properties [57]. Niranthin, the first lignan compound isolated from *P. niruri* [58], showed potent anti-HBV activity [49]. Four novel lignans, namely 5-demethoxyniranthin, urinatetralin, dextrobursehernin, and urinaligran, were identified in *P. urinaria* [59]. Additionally, nine common lignans were also identified. The result of the analysis, which included the quantification of the total lignans, indicated that *P. amarus* was the most abundant source of lignans, followed by *P. fraternus* and *P. debilis*. Notably, the plants' leaves contained the highest concentrations of lignans [60]. *P. amarus* has yielded numerous lignans, including the bitter component phyllanthin and the non-bitter component hypophyllanthin [61]. Additional lignans obtained from *P. amarus* comprise niranthin, phyltetralin, nirtetralin, isonirtetralin, hinokinin, lintetralin, isolintetralin, demethylenedioxy-niranthin, and 5-demethoxy-niranthin, among others [27]. 3-(3,4-dimethoxy-benzyl)-4-(7-methoxy-benzo [1,3]dioxol-5-yl-methyl)-dihydrofuran-2-one and 4-(3,4-dimethoxy-phenyl)-1-(7-methoxy-benzo [1,3]dioxol 5-yl)-2,3-bis-methoxymethyl-butan-1-ol are two novel lignans obtained from *P. amarus* leaves [62]. Table 2 presents an overview of the various pharmacological activities of lignans found in Phyllanthus genus, demonstrating their potential benefits for humans.

**Table 2.** Lignans found in the *Phyllanthus* genus and their respective pharmacological potencies.

| Species | Lignan | Potency | Refs. |
|---|---|---|---|
| *P. brasiliensis* | • justicidin B <br> • tuberculatin <br> • phyllanthostatin A <br> • 5-O-β-d-glucopyranosyljusticidin B <br> • cleistanthin | • against Zika virus | [63] |
| *P. franchetianus* | • phychetin A <br> • phychetin B <br> • phychetin C <br> • phychetin D | • protection against liver injury <br> • anti-inflammation | [64] |
| *P. amarus* | • demethyleneniranthin | • anti-hyperglycemic activity | [65] |
| *Phyllanthus* | • niranthin | • anxiolytic agent | [66] |
| *P. taxodiifolius* | • 4-O-(2′,3′,4′-tri-O-methyl-β-D-xylopyranosyl) | • anticancer | [67] |
| *P. amarus* | • diphyllin | • anti-inflammatory | [68] |
| *Phyllanthus* | • phyllanthin | • inhibits MOLT-4 leukemic cancer cells | [69] |
| *P. brasiliensis* | • 5-O-β-d-glucopyranosyljusticidin B <br> • arabelline <br> • 4-O-β-d-apiofuranosyl-(1‴→6″)-β-d-glucopyranosyldiphyllin <br> • cleistanthin B <br> • phyllanthostatin A <br> • tuberculatin <br> • justicidin B | • antiedematogenic <br> • anti-inflammatory <br> • antinociceptive | [70] |
| *P. amarus* | • 7′-oxocubebin dimethylether | • cytotoxic activity against HeLa cell line | [71] |

**Table 2.** *Cont.*

| Species | Lignan | Potency | Refs. |
|---|---|---|---|
| *P. songboiensis* | • arylnaphthalene lignan, (+)-acutissimalignan A | • cytotoxic toward HT-29 human colon cancer cells | [72] |
| *P. niruri* | • niranthin | • anti-hepatitis B virus | [58] |
| *P. flexuosus* | • phyllanthusmin C | • moderate <br> • cytotoxicity against the ECA109 human esophagus cancer cell line | [73] |
| *P. poilanei* | • arylnaphthalene lignan lactones | • cytotoxic toward HT-29 human colon cancer | [74] |
| *P. niruri* | • nirtetralin A <br> • nirtetralin B | • cells | [75] |
| *P. amarus* | • niranthin | • anti-hepatitis B virus | [76] |
| *P. niruri* | • phyllanthin <br> • hypophyllanthin <br> • phyltetralin <br> • niranthin | • anti-leishmanial agent | [77] |

A number of efforts have been allocated to enhancing lignan synthesis in *Phyllanthus*. In this review, we particularly highlighted the application of different elicitors as well as extraction methods in various *Phyllanthus* species (Table 3).

**Table 3.** Research on enhancing lignan synthesis in *Phyllanthus*.

| Species | Part of Plant | Method/Assay/Determination | Metabolite | Refs. |
|---|---|---|---|---|
| *P. amarus* | whole plants | • chloroform extraction <br> • LC-MS/GC-MS/In silico analysis | • phyllanthin <br> • niranthin <br> • corrilagin | [78] |
| *P. amarus* | shoot culture | • in vitro culture MS+Kinetin 0.25 mg/L <br> • methanolic extraction <br> • HPLC | • phyllanthin <br> • hypophyllanthin | [79] |
| *P. tenellus* | leaf derived callus | • in vitro culture MS+auxins+cytokinins <br> • methanolic extraction <br> • HPLC/LC-HRMS | • phyltetralin <br> • phyllanthin <br> • hypophyllanthin <br> • niranthin | [53] |
| *P. acuminatus* | hairy roots | • in vitro culture MS+Methyl jasmonate <br> • ethanolic extract <br> • LC-MS | • compound from phenylpropanoid pathway | [80] |
| *P. niruri* | aerial part | • methanolic extraction <br> • TLC | • phyllanthin | [81] |
| *P. niruri* | leaves | • acetone extraction <br> • fractionated using vacuum liquid chromatography (VLC) technique | • nirtetralin b <br> • phyllanthin | [82] |
| *P. amarus* | aerial part | • greenhouse experiment <br> • methanolic extract <br> • GC-MS | • 5-demethoxy-niranthin <br> • phyllanthin <br> • phyltetralin <br> • 5-demethoxy-nirtetralin <br> • nirtetralin <br> • niranthin | [83] |
| *P. amarus* | callus | • methanolic extract <br> • MS+phenylalanine+naphthalene acetic acid <br> • HPLC | • phyllanthin <br> • hypophyllanthin | [84] |
| *P. amarus* | callus from shoot | • MS+2.27 µM thidiazuron <br> • HPLC | • phyllanthin <br> • hypophyllanthin | [85] |

## 5. Endophytic Fungi and Lignan Biosynthesis

Several endophytic fungi produce lignans, such as arylnaphthalene lignan podophyllotoxin [13]. It has been reported that *Nigrospora* sp., an endophytic fungus derived from *P. amarus*, possesses the ability to synthesize antioxidant compounds, phyllanthin, and hypophyllanthin [48]. In addition, A. niger obtained from *P. amarus* leaves and exhibited

antihepatotoxic activity. Endophytes can also show the same therapeutic properties as their host plants and may also secrete known or novel metabolites [45]. For instance, endophytic *Streptomyces* sp. KCA1 yields the lignan 2,4-Di-tert-butylphenol, which is a promising candidate for the formulation of an antibacterial and anticancer compound [86]. Elicitation by the root endophytic fungus Piriformospora indica generated $H_2O_2$ molecules in the hairy roots of Linum album plants that led to changes in physiological, biochemical, and molecular responses, including changes in the antioxidant mechanism and synthesis of lignans [87]. The determination of metabolic contents in reaction to this elicitation indicates that both enzymatic and non-enzymatic defense mechanisms are significant in the adaptation of L. album hairy roots to *P. indica* elicitation.

Endophytic microbes support growth, supply nutrients, and increase stress tolerance in host plants. Strong activation of the immune system in plants under stress results in increased ROS production, which causes internal oxidative damage and death if the amount is excessive. Endophytes show the ability to mitigate abiotic stress through the modulation of local or systemic mechanisms. This modulation substantially increases leaf proline accumulation, chlorophyll content, and root activity. Inoculation with endophytic microbes also considerably increased the concentration of antioxidant enzymes, which are responsible for reducing ROS accumulation [88].

The function of endophytic organisms in promoting plant stress resistance via the Induced Systemic Resistance (ISR) mechanism has been thoroughly investigated. Endophytic microbes signal plants to increase the synthesis of jasmonic acid (JA). The JA signaling pathway is composed of two distinct branches: (1) the ERF-branch, determined by transcription factors of the ERF type, and (2) the MYC-branch, determined by transcription factors of the MYC type, including MYC2. Ethylene (ET) and the ERF-branch collaborate in order to stimulate the expression of defense-related genes, including PLANT DEFENSIN1.2 (PDF1.2), which enables plants to effectively fend off necrotrophic pathogens. Abscisic acid (ABA) and the MYC-branch collaborate to stimulate the expression of defense-related genes, including VEGETATIVE STORAGE PROTEIN2, which is responsible for directing insect defense. The equilibrium between the ERF-branch and the MYC-branch is contingent upon the concurrent activation of the ET response in the ET/ABA pathway and the interactions between JA and ABA that are triggered by MYC2/3/4 [89]. MYC2 is involved in the beneficial microbe-triggered ISR mechanism [90]. JA and ET are involved in ISR activation in response to elicitation by activating the MYC2 transcription factor, a key regulator of most JA signaling pathways. Elicitation by endophytic microbes in plant roots causes accumulation of JA and ET, which are then moved towards the distal parts of the plant through signals of phloem mobility. In conjunction with the presence of reactive oxygen species (ROS), this JA/ET signaling induces NPR-1 protein expression to be followed by the activation of genes associated with plant resistance. The JA signaling pathway activates the PAL1 gene, which encodes phenylalanine ammonia-lyase, a resistance-related gene involved in the regulation of phenylpropanoid precursor synthesis of lignans and other phytochemicals [91] (Figure 3).

Several endophytic microbes have been isolated from *P. niruri*, and their activity potential has been identified. An endophytic fungus from *P. niruri* identified as *Fusarium oxysporum* can produce indole acetic acid (IAA) [92]. Several endophytic actinobacteria isolated from *P. niruri* have been shown to tolerate a wide range of pH, temperature, and salinity and to produce siderophores that induce plant growth and have antimicrobial and antioxidant properties [93]. The endophytic microorganisms that were obtained from *P. amarus*, namely *Acinetobacter* sp. and *Bacillus* sp., exhibited tolerance to salinity and enhanced various plant growth parameters including phosphate content, germination, and seed vigor index [94]. Isolated from *P. niruri*, *E. gomezpompae* produced phyllanthin during its 21-day growth in Potato Dextrose Broth (PDB) medium [46].

Fungal extracts have been effectively exploited to enhance lignan synthesis in cell suspensions of *Linum* species. *P. indica* effectively elicits lignan production in hairy root

cultures of *L. album*, providing plants with antioxidant machinery [95]. An extract of *Rhizopus tolonifera* enhanced the accumulation of lariciresinol in *L. album* cell cultures [96].

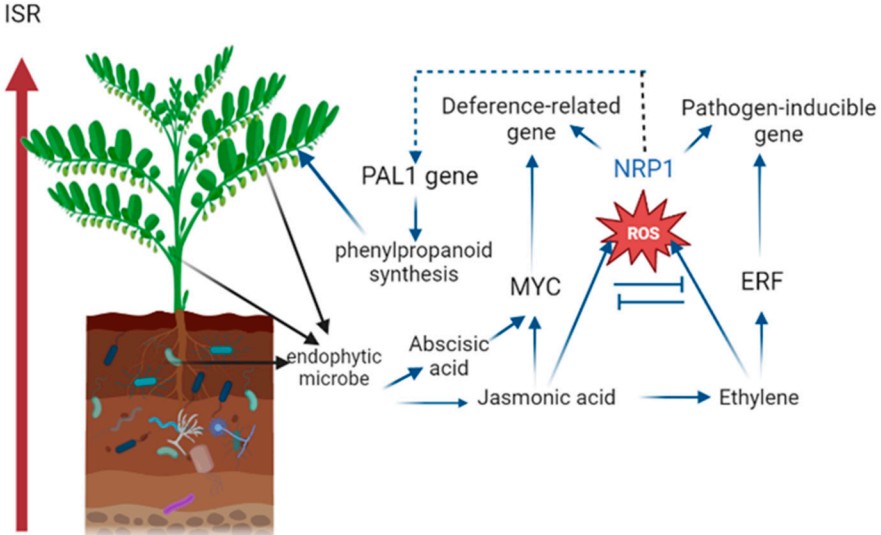

**Figure 3.** The mechanism of Induced Systemic Resistance (ISR) elicitation by endophytic microbes [89–91].

## 6. Drought Stress and Lignan Biosynthesis

Drought stress is hypothesized to change the physiology and secondary metabolite production of medicinal plants, including *Phyllanthus*. The production of amino acids (proline, GABA, alanine, and valine), malic acid, succinic acid, and citric acid, as well as glucose and corilagin, increased substantially in *P. niruri* plants subjected to drought stress [97]. Gas chromatography–mass spectrophotometry (GC-MS) analysis revealed that *P. niruri* induced different bioactive compounds during drought stress treatments [98]. The drought treatment also stimulated genes involved in the biosynthesis of amino acids, fatty acids, isoprenoid, and shikimate, among others, in the metabolic pathways of secondary metabolites in *P. urinaria*. Water stress led to an upregulation in the expression of twelve enzymes that are involved in the biosynthesis of phenylpropanoids. The enzymes that are involved in the aforementioned processes are glutathione peroxidase, cinnamoyl CoA reductase, cinnamyl-alcohol dehydrogenase, shikimate dehydrogenase, trans-cinnamate 4-monooxygenase, transketolase, and transaldolase, Class III peroxidase, and caffeate O-methyltransferase (COMT-1). In contrast, other enzymes in this pathway exhibited downregulation following drought stress including PAL, F5H, alpha/beta-D-glucosidase, caffeic acid O-/3-O-methyltransferase, caffeoyl-CoA-O-methyltransferase, 3-phosphoshikimate 1-carboxyvinyltransferase, and coumarate-3-hydroxylase. During water stress, 4-Coumarate CoA ligase expression increased, whereas caffeine synthase expression remained unchanged [99]. These results indicate that drought stress can lead to increased production of phenylpropanoids.

Drought stress induces signaling pathways through two mechanisms: ABA synthesis and JA synthesis. The activation of the 9-cis-epoxycarotenoid dioxygenase (NCED) gene cluster induces the ABA response, resulting in an elevation of ABA concentrations [100]. When ABA levels increase in plants under drought stress, receptors recognize ABA and activate SnRK2. SnRK2 activation causes downstream target phosphorylation events and triggers physiological and molecular responses such as germination, stomatal regulation, root development, photosynthesis, and regulation of stress-related genes to achieve drought tolerance [101].

Genes including allene oxide synthase (AOS), allene oxide cyclase (AOC), lipoxygenases (LOX), and OPDA reductase 3 (OPR3) are activated to produce the JA response [102]. JA plays a role prior to ABA synthesis, with elevated concentrations leading to the degra-

dation of the jasmonate ZIM-domain (JAZ) protein. This degradation, in turn, stimulates the expression of stress response-related gene transcription factors, including MYC2. Furthermore, these hormones induce plant drought tolerance, including ROS binding and stomatal regulation [100]. Elevated concentrations of ABA and JA activate MYC2, which is important in the regulation of crosstalk between the JA signaling pathway and various phytohormone pathways, including the ABA, SA, GA, and auxin pathways [90]. SnRK and MYC2 activated by the JA and ABA signaling pathway induce the expression of drought tolerance-related genes such as calmodulin, disease-resistance protein RPM1, CAT, SOD, and HSP70. The RPM1 gene encodes a disease-resistance protein involved in calcium signaling pathways and stress tolerance. Other genes are crucial in regulating tolerance to oxidative stress. The gene encoding sucrose synthase is regulated by ABA and JA levels, resulting in increased accumulation of sucrose and amino acids in response to drought stress. All of these mechanisms balance plant biomass accumulation with the production of antioxidant enzymes so plants can survive drought conditions [103]. Amino acids function as osmolytes to balance the osmotic potential and as binders of ROS during drought conditions. Aromatic amino acids are precursors for several secondary metabolites important for plant growth. Key phenylpropanoid pathway enzymes, such as PAL, 4-coumaroyl CoA ligase (4CL), and C4H, are involved in plant stress responses [104]. The enzyme 4-coumarate-CoA ligase 3 (4CL3), activated by the transcription factor WRKY34, is the primary rate-limiting enzyme of lariciresinol biosynthesis. WRKY 34 is a transcription factor with pleiotropic effects, including lignan biosynthesis and stress tolerance [43] (Figure 4).

## 7. Synergetic Treatment to Increase Lignan Production

Based on the known signaling responses to drought stress and elicitation by endophytic microbes, several hypotheses have been developed to explain how the crosstalk between endophytic microbes and drought stress responses regulates lignan production. Initially, specific receptors or binding sites for elicitors detect both drought stress and endophytic microbial presence. Upon recognition, these receptors trigger NADPH oxidase activity, resulting in a $Ca^{2+}$ burst in the cytoplasm, which stimulates cytoplasmic acidification, ROS discharge, and activation of MAPK and G-protein. The engagement of G-proteins causes stimulation of phospholipases and ion channels, which in turn stimulates the levels of cAMP, IP3, and DAG, as well as the target kinases PKA and PKC. This cascade ultimately enhances MAPK phosphorylase activity, which initiates hormone signaling and expression of genes, leading to enzyme synthesis.

Furthermore, NCED gene activation occurs in response to drought stress signaling so that ABA levels increase and the SnRK2 protein is activated. Endophytic microbial signaling increases JA levels, which further stimulates ABA levels. During the signal transduction process, crosstalk between ABA is activated by JA signaling and ET signaling in the ISR mechanism. ABA and JA activate the MYC2 transcription factor, while ET activation activates the NPR1 gene, which also activates the MYC2. Activation of SnRK2, MYC2, and the NPR1 gene activates stress-related genes such as the PAL gene, which is involved in the phenylpropanoid pathway and leads to increased lignan synthesis. MAPK phosphorylation activates transcription factors such as WRKY34, which can increase 4CL3 gene expression. In contrast, ABA synthesis activates a $Ca^{2+}$ burst so that the MYB2 and ABRE transcription factors, which are both PLR gene promoters, are activated, causing PLR gene expression to increase. The 4CL3 and PLR genes play a role in the conversions of pinoresinol to lariciresinol and secoisolariciresinol, which is hypothesized to be a phyllanthin precursor [33,43,105,106] (Figure 5). The non-coding sequences of the Lu-PLR gene have the same nucleotide motif as the non-coding sequences of MYB1, a transcription factor for phenylpropanoid biosynthesis genes; WRKY, a transcription factor for elicitor response genes; W box, a group of cis-regulatory elements related to activation of gene expression due to injury; ABRE and E box, which are cis-regulatory elements associated with ABA regulation; and MYB2, a transcription factor related to gene expression induced by ABA and drought [107].

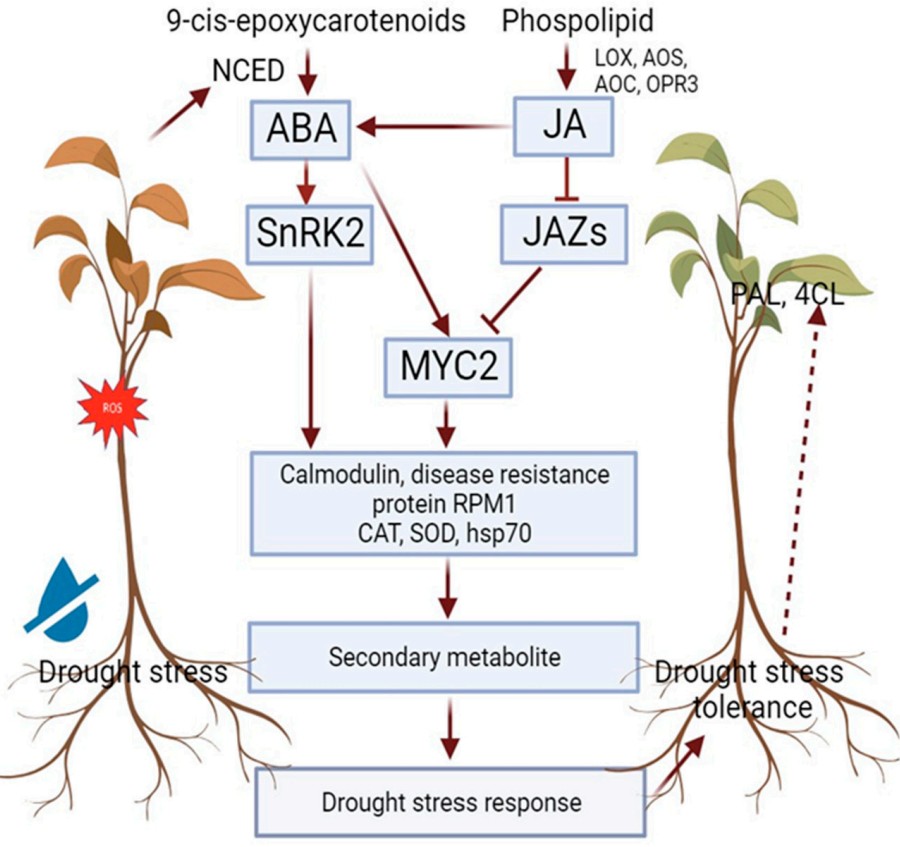

**Figure 4.** Induction of drought stress tolerance in ABA and JA signaling pathways [90,100–104].

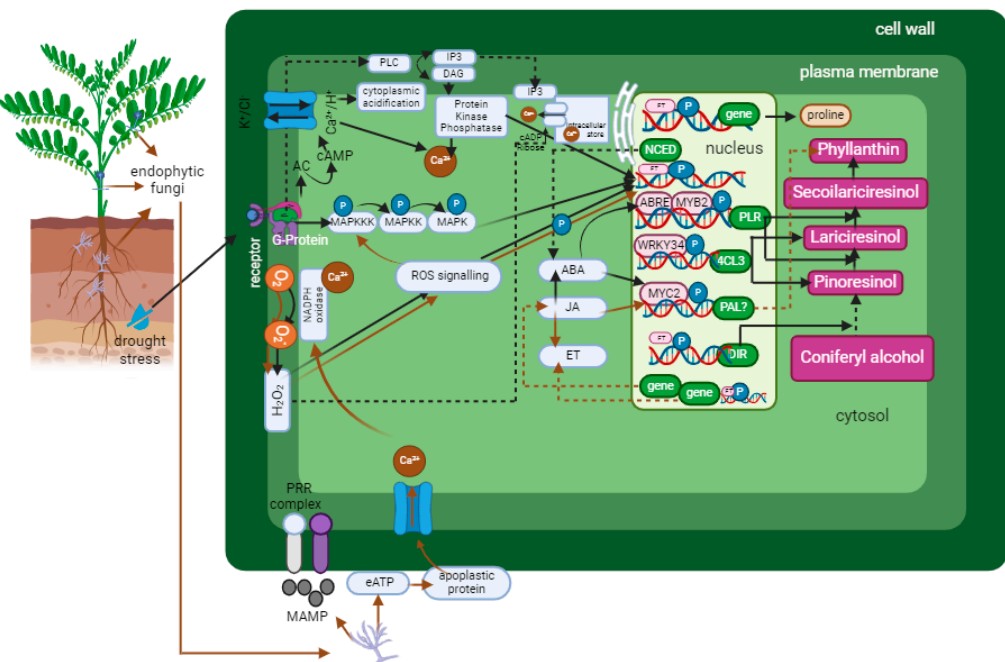

**Figure 5.** An overview of the hypothesized mechanisms by which endophytic fungus elicitation and drought stress increase lignan synthesis.

Synergistic elicitation, a strategy aimed at boosting the generation of plant secondary metabolites like lignans, involves the use of various elicitors together. This method, which often includes the use of elicitors, such as methyl jasmonate, salicylic acid, and yeast extract, can significantly enhance the production of desired compounds. Research indicates

that using a combination of elicitors, particularly yeast extract, not only increases lignan accumulation but also does so without harming plant biomass [108,109]. This approach leverages the complementary effects of different elicitors to maximize the induction of the biosynthetic pathways responsible for lignan production. The use of synergistic elicitation can be a promising strategy for improving the biotechnological production of lignans and other valuable secondary metabolites in plant cell cultures [110]. This review provides a new insight into the concept of elicitation on medicinal plants for enhancing potentially pharmacological compounds, especially lignan, through endophyte and host interactions as well as drought treatment. The interaction between endophytes and plants is informed by strong synergy, as endophytes are the primary drivers of the host plant's resistance to abiotic and biotic stress [111]. Understanding the balance between plant defense and microbial virulence may offer better insights into the potential of chemical elicitors to influence the full capacity of endophytes. Combining different elicitors, such as endophyte microbes and drought stress, can lead to a more significant increase in the synthesis of target compounds. This approach leverages the complementary effects of different elicitors to maximize the induction of the biosynthetic pathways responsible for producing target compounds. In summary, synergistic elicitation using endophyte microbes and drought stress can be a promising strategy for enhancing the production of bioactive compounds in plants. By combining different elicitors, it is possible to maximize the induction of the biosynthetic pathways responsible for producing target compounds, leading to improved plant growth and secondary metabolite production. The weakness of synergistic elicitation between endophyte microbes and drought stress lies in the complexity of the interactions and the challenges associated with practical implementation. While the combination of endophytes and drought stress can enhance the production of bioactive compounds in plants, several factors need to be considered such as the specific plant–microbe interactions, environmental variability, and the complex mechanisms involved [112,113].

## 8. Enhancing Lignan Production in *Phyllanthus* through Omics Approach

The omics-based approach for studying the synergistic elicitation of endophyte microbes and drought stress on plants employs various multi-omics technologies to analyze the interaction between plants and endophytes under different abiotic stress conditions. This method aims to bridge existing gaps in understanding how physiological growth of plants is influenced by endophytes and to enhance insights into the molecular and biochemical processes driving the interactions among endophytes, plants, and their surrounding environment [114]. By integrating technologies such as transcriptomics, proteomics, metabolomics, and phenomics, researchers can gain a more holistic view of the intricate dynamics between endophytes and plants when subjected to drought stress (Table 4).

Lignan omics research faces limitations due to its complexity and the heterogeneity in plant secondary metabolites. Integrated multi-omics approaches like genomics, transcriptomics, proteomics, and metabolomics are used to overcome these limitations but may not fully capture the complexity of lignan and its regulation as they struggle to identify low-abundance proteins. Research on the treatment of biotic and abiotic elicitation to enhance lignan synthesis has been conducted by several researchers. The omics approach is becoming the tool to identify the effectiveness of this treatment. Current research on the synergistic elicitation of endophyte fungi and drought exposure to enhance lignan synthesis, especially in medicinal plants from the genus *Phyllanthus,* is still limited.

**Table 4.** Omics research on elicitation and enhancing lignan production in plants.

| Species | Elicitation Treatment | Lignan Enhancement | Omics Approach | Refs. |
|---------|----------------------|--------------------|----------------|-------|
| *Glycine max* | Co-cultivation with endophyte *Piriformospora indica* | • lignin biosynthesis-related gene<br>• phenylpropanoid | Trancriptomics | [115] |
| *Isatis indigotica* | Hairy root culture under salt and drought stresses | • IiWRKY34 increasing lignan biosynthesis | Trancriptomics | [43] |
| *Isatis indigotica* | Hairy root culture with salicylic acid | • Ii049, a transcription factor from the AP2/ERF family, increases lignan biosynthesis | Genomics | [116] |
| *Sinopodophyllum hexandrum* | Water-deficit treatment | • the phenylpropanoid pathway and lignan podophyllotoxin biosynthesis: PAL, 4CL, PLR<br>• podophyllotoxin | Genomics | [117] |
| *Adianum nelumboides* | Drought stress treatment | • lignan biosynthesis | Metabolomics Transcriptomics | [118] |
| *Tamarix taklamakanensis* | Drought stress treatment | • lignan biosynthesis | Metabolomics Transcriptomics | [119] |
| *Taxus baccata* | Salicylic acid | • phenylpropanoid biosynthetic | Metabolomics | [120] |
| *Bryophyllum* sp. | Methyl jasmonate Salicylic acid | • lignan biosynthesis | Metabolomics | [121] |
| *Phyllanthus acuminatus* | Methyl jasmonate Salicylic acid | • phenylpropanoid biosynthetic | Metabolomics | [80] |
| *Cicer arietinum* | Salicylic acid Chitosan Hydrogen peroxide | • matairesinol<br>• secoisolariciresinol | Metabolomics | [122] |
| *Castilleja tenuiflora* | Salicylic acid Hydrogen peroxide | • tenuifloroside<br>• eudesmin<br>• magnolin, kobusin<br>• sesamin | Metabolomics | [123] |
| *Haplophyllum virgatum* Spach. | chitin treatment | • phenylpropanoid biosynthesis-related gene<br>• podophyllotoxin | Genomics Metabolomics | [124] |
| *Linum lewisii* | Methyl jasmonate | • justicidin B | Genomics Metabolomics | [125] |
| *Linum usitatissimum* L. | Chitosan oligosaccharides | • (neo)lignans biosynthesis | Metabolomics | [126] |

## 9. Conclusions and Future Perspective

Lignans are essential in plant–environmental interactions and crucial in abiotic stress responses. Abiotic stress, such as drought, might negatively impact *Phyllanthus* growth, production, and yield. Plants have evolved adaptive mechanisms to cope with the stresses, but they still face significant losses in performance and productivity under drought exposure. These conditions cause the emergence of oxidative stress, leading to the accumulation of ROS. Some endophyte fungi can potentially be an elicitor to enhance lignan synthesis in medicinal plants. The medicinal plant–endophyte relationship helps the production of secondary metabolites, which is crucial for human health and can be optimized under in vitro and in vivo conditions. By modulating the biosynthesis of secondary metabolites,

it is possible to enhance the production of lignans, thereby improving *Phyllanthus* species. Future research, incorporating a multidisciplinary approach and leveraging advanced metabolomics platforms, is crucial for a deeper understanding of how lignan impacts the symbiotic relationships between medicinal plants and their endophyte fungi, particularly within the Phyllanthus genus. Metabolomics enables the quantification of previously unidentified metabolites following plant–endophyte treatment, making multi-omics analyses an increasingly important tool for predicting and enhancing medicinal plant–endophyte interactions to boost lignan synthesis.

**Author Contributions:** W.N. conducted primary literature research, outlined the topics, and wrote the manuscript; L.D., R.R.E. and A.F. provided directions and determined the theme for the review; A.F. and R.R.E. validated the content, and reviewed and edited the manuscript. All authors have read and agreed to the published version of the manuscript.

**Funding:** This research was funded by an ITB research grant under the scheme of excellent research program 2022 (Contract No. 293/IT1.B07.1/TA.00/2022).

**Data Availability Statement:** Not applicable.

**Conflicts of Interest:** The authors declare no conflicts of interest.

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
