# Peer review of "Phyllanthus Lignans: A Review of Biological Activity and Elicitation"

_horticulturae, doi:10.3390/horticulturae10020195_

Round 1
Reviewer 1 Report
Comments and Suggestions for Authors
You can see all in manuscript

Check in manuscript
Author Response
Please find our response in attachment

Reviewer 2 Report
Comments and Suggestions for Authors
The review by Nawfetrias and co-authors reports the current knowledge on the potency of medicinal lignans produced by Phyllanthus and how lignan synthesis could be increased through biotic and abiotic elicitation. In general, the manuscript is well written, and the review could be published after minor revisions:
1) Please add a small paragraph about the modern chromatographic, spectroscopic, and spectrometric methods used to purify and identify Lignans in the general introduction. It is important because the words LC-MS are reported in the text, and considering that a majority of the readers of the journal will probably not be chemistry experts, reporting this information and appropriate literature could help them find important information.
2) Please add a Table, as supplementary material, reporting the name, molecular formula, structure and literature of all the Lignans isolated from the Phyllanthus genus. This is important to visualize the structure of the metabolites and summarize the information for the readers.
Author Response
Please find our response in attachment

Reviewer 3 Report
Comments and Suggestions for Authors
Dear Authors
Below are the observations and comments made on your manuscript, please take them into account when updating it.
1) In a figure or figures, include the structural nuclei for the lignan family and its molecules (Basic skeleton structure of lignans and their classes).
2) You must enter the databases and date range for which you performed the search, including the search key(s) to capture the information.
3) Include botanical description of the family, genus and some species.
4) Could you include photos of some species?
5) Include a table listing the following information by column: a) Plant (species), b) Part used, c) Method / Assay / Determination, d) Activity / Effect, e) Metabolite, f) Reference
6) In the manuscript indicate why and for what purpose you conducted this review. Also include the strengths and weaknesses of this review compared to others.
Authors to make changes and update the manuscript
Kind regards
Reviewer
Author Response
Please find our response in attachment

Reviewer 4 Report
Comments and Suggestions for Authors
The authors have addressed an interesting review topic on Phyllanthus lignans and their biological activity and elicitation. However, after reading the manuscript, I noticed that, except for a brief mention in the conclusion, the authors did not address molecular and biochemical omic approaches in the main text. Therefore, as a major recommendation, I suggest that the authors provide the current panorama or their perspectives in the text (not only in the conclusion) about all the omic approaches (proteomics, transcriptomics, metabolomics, etc.) related to the present research topic, providing adequate tables and figures. Moreover, the authors could include tables in the manuscript to make it more informative to the readers. Furthermore, I recommend citing more articles published in 2023 and 2024 as well as more recent articles throughout the text.
Author Response
Please find our response in attachment

Round 2
Reviewer 3 Report
Comments and Suggestions for Authors
Authors: The manuscript has been updated according to the comments made during the revision.
Regards. Reviewer
Reviewer 4 Report
Comments and Suggestions for Authors
I have no more comments.